# Inequalities on General $L_p$-Mixed Chord Integral Difference

Hongying Xiao [1], Weidong Wang [2,*] and Zhaofeng Li [2,*]

[1] Faculty of Science, Yibin University, Yibin 644000, China; xiaohongying715@163.com
[2] Department of Mathematics, China Three Gorges University, Yichang 443002, China
* Correspondence: Wangwd722@163.com (W.W.); kelly0128@163.com (Z.L.)

**Abstract:** In this article, we introduce the concept of general $L_p$-mixed chord integral difference of star bodies. Further, we establish the Brunn–Minkowski type, Aleksandrov–Fenchel type and cyclic inequalities for the $L_p$-mixed chord integral difference.

**Keywords:** general $L_p$-mixed chord integral difference; volume difference; $L_p$-radial Minkowski combination; $L_p$-radial Blaschke Minkowski homomorphism

## 1. Introduction

The setting for this paper is $n$-dimensional Euclidean spaces $\mathbb{R}^n$ ($n \geq 1$). Let $K$ and $L$ be two convex bodies (compact, convex subsets with nonempty interiors) in $\mathbb{R}^n$. $V$ denotes the volume. If $K$ is a compact star-shaped (about the origin) set in $\mathbb{R}^n$, then its radial function, $\rho_K = \rho(K, \cdot) : \mathbb{R}^n \setminus \{0\} \to [0, \infty)$, is defined by (see [1]):

$$\rho(K, u) = \max\{\lambda \geq 0, \lambda u \in K\}, \ u \in \mathcal{S}^{n-1}.$$

If $\rho_K$ is positive and continuous, $K$ is called a star body (about the origin), and $S^n$ denotes the set of star bodies in $\mathbb{R}^n$. $S_0^n$ is the subset of $S^n$ containing the origin in their interiors. The unit sphere in $\mathbb{R}^n$ is denoted by $\mathcal{S}^{n-1}$, and $B$ denotes the standard unit ball in $\mathbb{R}^n$.

The classical Brunn–Minkowski inequality is (see [2])

$$V(K + L)^{\frac{1}{n}} \geq V(K)^{\frac{1}{n}} + V(L)^{\frac{1}{n}},$$

where $+$ denotes vector or the Minkowski sum of two sets, i.e., $A + B = \{a + b : a \in A, b \in B\}$.

In 2004, Leng (see [3]) presented a new generalization of the Brunn–Minkowski inequality for the volume difference of convex bodies.

**Theorem 1.** *Suppose that $K, L$ and $D$ are compact domains, and $D \subset K, D' \subset L$, $D'$ is a homothetic copy of $D$. Then*

$$[V(K + L) - V(D + D')]^{\frac{1}{n}} \geq [V(K) - V(D)]^{\frac{1}{n}} + [V(L) - V(D')]^{\frac{1}{n}}.$$

*The equality holds if and only if $K$ and $L$ are homothetic and $(V(K), V(D)) = \mu(V(L), V(D'))$, where $\mu$ is a constant.*

Leng's result is a major extension of the classical Brunn–Minkowski inequality and attracts more and more attention (see [4–6]).

In 1977, Lutwak introduced the notion of a mixed width-integral of convex bodies (see [7]), and the dual notion, mixed chord-integrals of star bodies was defined by Lu (see [8]). Later, as a part of the asymmetric $L_p$ Brunn–Minkowski theory, which has its

origins in the work of Ludwig, Haberl and Schuster (see [9–13]), Feng and Wang generalized the mixed chord-integrals to general mixed chord-integrals of star bodies (see [14]). For $K_1, \cdots, K_n \in S_0^n$ and $\tau \in (-1, 1)$, the general mixed chord-integral $C^{(\tau)}(K_1, \cdots, K_n)$ is defined by

$$C^{(\tau)}(K_1, \cdots, K_n) = \frac{1}{n} \int_{\mathcal{S}^{n-1}} c^{(\tau)}(K_1, u) \cdots c^{(\tau)}(K_n, u) du,$$

here, $c^{(\tau)}(K, \cdot) = f_1(\tau)\rho(K, \cdot) + f_2(\tau)\rho(-K, \cdot)$, and the functions $f_1(\tau)$ and $f_2(\tau)$ are defined as follows

$$f_1(\tau) = \frac{(1+\tau)^2}{2(1+\tau^2)}, \quad f_2(\tau) = \frac{(1-\tau)^2}{2(1+\tau^2)}.$$

In 2016, Li and Wang extended the general mixed chord-integral to the general $L_p$-mixed chord integral of star bodies (see [15]): For $K_1, \cdots, K_n \in S_0^n$, $p > 0$ and $\tau \in (-1, 1)$, the general $L_p$-mixed chord integral $C_p^{(\tau)}(K_1, \cdots, K_n)$ of $K_1, \cdots, K_n$ is defined by

$$C_p^{(\tau)}(K_1, \cdots, K_n) = \frac{1}{n} \int_{\mathcal{S}^{n-1}} c_p^{(\tau)}(K_1, u) \cdots c_p^{(\tau)}(K_n, u) du. \tag{1a}$$

Here, $c_p^{(\tau)}(K, \cdot)$ is defined by

$$c_p^{(\tau)}(K, u) = \left( f_1(\tau)\rho^p(K, u) + f_2(\tau)\rho^p(-K, u) \right)^{\frac{1}{p}},$$

for any $u \in \mathcal{S}^{n-1}$, and $f_1(\tau)$ and $f_2(\tau)$ are chosen as (see [16])

$$f_1(\tau) = \frac{(1+\tau)^p}{(1+\tau)^p + (1-\tau)^p}, \quad f_2(\tau) = \frac{(1-\tau)^p}{(1+\tau)^p + (1-\tau)^p}.$$

Obviously, $f_1(\tau)$ and $f_2(\tau)$ satisfy

$$f_1(\tau) + f_2(\tau) = 1,$$

$$f_1(-\tau) = f_2(\tau), \quad f_2(-\tau) = f_1(\tau).$$

$C_{p,i}^{(\tau)}(K, L)$ denotes that $K$ appears $n - i$ times, and $L$ appears $i$ times, which is

$$C_{p,i}^{(\tau)}(K, L) = \frac{1}{n} \int_{\mathcal{S}^{n-1}} c_p^{(\tau)}(K, u)^{n-i} c_p^{(\tau)}(L, u)^i du.$$

If constants $\lambda_1, \cdots, \lambda_n > 0$ exist such that $\lambda_1 c_p^{(\tau)}(K_1, u) = \cdots = \lambda_n c_p^{(\tau)}(K_n, u)$ for all $u \in \mathcal{S}^{n-1}$, star bodies $K_1, \cdots, K_n$ are said to have a similar general $L_p$-chord. For this general $L_p$-chord integral, Li and Wang gave the following inequalities (see [15]).

**Theorem 2.** *If $K, L \in S_o^n$ and $\tau \in (-1, 1)$, $p > 0$, then for $i \leq n - p$,*

$$C_{p,i}^{(\tau)}(K \tilde{+}_p L)^{\frac{p}{n-i}} \leq C_{p,i}^{(\tau)}(K)^{\frac{p}{n-i}} + C_{p,i}^{(\tau)}(L)^{\frac{p}{n-i}}, \tag{1b}$$

*for $n - p < i < n$ or $i > n$,*

$$C_{p,i}^{(\tau)}(K \tilde{+}_p L)^{\frac{p}{n-i}} \geq C_{p,i}^{(\tau)}(K)^{\frac{p}{n-i}} + C_{p,i}^{(\tau)}(L)^{\frac{p}{n-i}}, \tag{1c}$$

*with equality in each inequality if and only if $K$ and $L$ have a similar general $L_p$-chord. Here and in the following Theorems, $K \tilde{+}_p L$ denotes the $L_p$-radial Minkowski combination of $K$ and $L$.*

**Theorem 3.** *If $K_1, \cdots, K_n \in S_o^n$ and $\tau \in (-1, 1)$, $p > 0$, then for $1 < m \leq n$,*

$$C_p^{(\tau)}(K_1, \cdots, K_n)^m \leq \prod_{i=1}^{m} C_p^{(\tau)}(K_1, \cdots, K_{n-m}, K_{n-i+1}, K_{n-i+1}, \cdots, K_{n-i+1}), \quad (1d)$$

*with equality if and only if $K_{n-m+1}, \cdots, K_n$ all have a similar general $L_p$-chord.*

**Theorem 4.** *If $K, L \in S_o^n$ and $\tau \in (-1, 1)$, $p > 0$, then for $i < j < k$,*

$$C_{p,j}^{(\tau)}(K, L)^{k-i} \leq C_{p,i}^{(\tau)}(K, L)^{k-j} C_{p,k}^{(\tau)}(K, L)^{j-i}, \quad (1e)$$

*with equality if and only if $K$ and $L$ have a similar general $L_p$-chord.*

## 2. Main Results

Inspired by Leng's idea, this article deals with the general $L_p$-chord integral of star bodies and gives some inequalities for the general $L_p$-chord integral difference.

**Theorem 5.** *Let $K, L, M, M' \in S_o^n$ and $\tau \in (-1, 1)$, $p > 0$. If $K$ and $L$ have similar general $L_p$-chord and $M \subseteq K$, $M' \subseteq L$, then for $i \leq n - p$,*

$$[C_{p,i}^{(\tau)}(K \tilde{+}_p L) - C_{p,i}^{(\tau)}(M \tilde{+}_p M')]^{\frac{p}{n-i}} \geq [C_{p,i}^{(\tau)}(K) - C_{p,i}^{(\tau)}(M)]^{\frac{p}{n-i}} + [C_{p,i}^{(\tau)}(L) - C_{p,i}^{(\tau)}(M')]^{\frac{p}{n-i}}, \quad (1f)$$

*and for $n - p < i < n$ or $i > n$,*

$$[C_{p,i}^{(\tau)}(K \tilde{+}_p L) - C_{p,i}^{(\tau)}(M \tilde{+}_p M')]^{\frac{p}{n-i}} \leq [C_{p,i}^{(\tau)}(K) - C_{p,i}^{(\tau)}(M)]^{\frac{p}{n-i}} + [C_{p,i}^{(\tau)}(L) - C_{p,i}^{(\tau)}(M')]^{\frac{p}{n-i}}, \quad (1g)$$

*with equality in each inequality if and only if $M$ and $M'$ have a similar general $L_p$-chord.*

**Theorem 6.** *Let $K_1, \cdots, K_n$ and $M_1, \cdots, M_n \in S_o^n$, and $\tau \in (-1, 1)$, $p > 0$. If $M_i \subseteq K_i$, $i = 1, 2, \cdots, n$, $K_1, \cdots K_n$ have similar general $L_p$-chord, then for $1 < m \leq n$,*

$$[C_p^{(\tau)}(K_1, \cdots, K_n) - C_p^{(\tau)}(M_1, \cdots, M_n)]^m \geq$$

$$\prod_{i=1}^{m} [C_p^{(\tau)}(K_1, \cdots, K_{n-m}, K_{n-i+1}, K_{n-i+1}, \cdots, K_{n-i+1}) - C_p^{(\tau)}(M_1, \cdots, M_{n-m}, K_{n-i+1}, M_{n-i+1}, \cdots, M_{n-i+1})], \quad (1h)$$

*with equality if and only if $M_1, \cdots, M_n$ all have a similar general $L_p$-chord.*

**Theorem 7.** *Let $K, L, M, M' \in S_o^n$ and $\tau \in (-1, 1)$, $p > 0$. If $K$ and $L$ have similar general $L_p$-chord, then for $i < j < k$,*

$$[C_{p,j}^{(\tau)}(K, L) - C_{p,j}^{(\tau)}(M, M')]^{k-i} \geq [C_{p,i}^{(\tau)}(K, L) - C_{p,i}^{(\tau)}(M, M')]^{k-j} [C_{p,k}^{(\tau)}(K, L) - C_{p,k}^{(\tau)}(M, M')]^{j-i}, \quad (1i)$$

*with equality if and only if $K$ and $L$ have a similar general $L_p$-chord.*

## 3. Preliminaries

For $K, L \in S^n$, the radial Blaschke linear combination $K \check{+} L$ and the radial Minkowski linear combination are defined by Lutwak (see [17]), respectively:

$$\rho(K \check{+} L, u)^{n-1} = \rho(K, u)^{n-1} + \rho(L, u)^{n-1}, \quad (2a)$$

and

$$\rho(K \tilde{+} L, u) = \rho(K, u) + \rho(L, u). \quad (2b)$$

In 2007, Schuster introduced the notion of radial Blaschke–Minkowski homomorphism (see [18–22]) as follows.

**Definition 1.** *A map $\Psi : S^n \to S^n$ is called a radial Blaschke–Minkowski homomorphism if it satisfies the following conditions:*

(1) $\Psi$ *is coninuous;*
(2) $\Psi$ *is radial Blaschke Minkowski additive, i.e.,* $\Psi(K \check{+} L) = \Psi K \tilde{+} \Psi L$ *for all $K, L \in S^n$;*
(3) $\Psi$ *intertwines rotations, i.e.,* $\Psi(\phi K) = \phi \Psi K,$ *for all $\phi \in SO(n)$ and $K \in S^n$.*

Here, $\Psi K \tilde{+} \Psi L$ denotes the radial sum of $\Psi K$ and $\Psi L$, and $K \check{+} L$ is the radial Blaschke sum of the star bodies $K$ and $L$.

In 2011, Wang et al. (see [23]) extended the notion of radial Blaschke–Minkowski homomorphism to $L_p$-radial Minkowski homomorphism as follows.

**Definition 2.** *A map $\Psi_p : S^n \to S^n$ is called an $L_p$-radial Minkowski homomorphism if it satisfies the following conditions:*

(1) $\Psi_p$ *is coninuous;*
(2) $\Psi_p$ *is radial Minkowski additive, i.e.,* $\Psi_p(K \tilde{+}_{n-p} L) = \Psi_p K \tilde{+}_p \Psi_p L$ *for all $K, L \in S^n$;*
(3) $\Psi_p$ *intertwines rotations, i.e.,* $\Psi_p(\phi K) = \phi \Psi_p K,$ *for all $\phi_p \in SO(n)$ and $K \in S^n$.*

Here, $\Psi_p K \tilde{+}_{n-p} \Psi_p L$ denotes the $L_{n-p}$ radial sum of $\Psi_p K$ and $\Psi_p L$, i.e., (see [9,24])

$$\rho(\Psi_p K \tilde{+}_{n-p} \Psi_p L, u)^{n-p} = \rho(\Psi_p K, u)^{n-p} + \rho(\Psi_p L, u)^{n-p}. \tag{2c}$$

For $0 < p < n$, the $L_p$-radial Blaschke linear combination $K \check{+}_p L$ was defined by Wang (see [25]):

$$\rho(K \check{+}_p L, u)^{n-p} = \rho(K, u)^{n-p} + \rho(L, u)^{n-p}. \tag{2d}$$

From Equations (2c) and (2d), we easily obtain

$$K \tilde{+}_{n-p} L = K \check{+}_p L. \tag{2e}$$

Here, we recall a special $L_p$-radial Minkowski homomorphism. In 2007, Yu, Wu and Leng (see [26]) introduced the quasi-$L_p$ intersection body $I_p K$ of a star body. Let $K$ be a star body in $\mathbb{R}^n$, then the quasi-$L_p$ intersection body $I_p K$ of $K$ is defined by:

$$\rho(I_p K, u)^p = \int_{\mathcal{S}^{n-1} \cap u^\perp} \rho(K, u)^{n-p} du.$$

Further, Wang (see [23]) proved that the operator $I_p : S^n \to S^n$ has the following properties: (1) $I_p$ is continuous with respect to radial metric; (2) $I_p(K \tilde{+}_{n-p} L) = I_p K \tilde{+}_p I_p L$ for all $K, L \in S^n$; (3) $I_p$ intertwines rotations, i.e., $\Psi_p(\phi K) = \phi \Psi_p K,$ for all $\phi_p \in SO(n)$ and $K \in S^n$, which means that the operator $I_p$ is a special $L_p$-radial Minkowski homomorphism.

Now, we list three Lemmas useful in the proof of Theorems 5–7.

In 1997, Losonczi and Páles (see [27]) extended Bellman's inequality as follows:

**Lemma 1.** *Let $a = \{a_1, a_2, \cdots, a_n\}$ and $b = \{b_1, b_2, \cdots, b_n\}$ ($n \geq 1$) be two sequences of positive real numbers and $p > 1$ such that $a_1^p - \Sigma_{i=2}^n a_i^p > 0$ and $b_1^p - \Sigma_{i=2}^n b_i^p > 0$. Then*

$$\left(a_1^p - \Sigma_{i=2}^n a_i^p\right)^{\frac{1}{p}} + \left(b_1^p - \Sigma_{i=2}^n b_i^p\right)^{\frac{1}{p}} \leq \left((a_1 + b_1)^p - \Sigma_{i=2}^n (a_i + b_i)^p\right)^{\frac{1}{p}}, \tag{2f}$$

*If $p < 0$ or $0 < p < 1$, then*

$$\left((a_1^p - \Sigma_{i=2}^n a_i^p)^{\frac{1}{p}} + (b_1^p - \Sigma_{i=2}^n b_i^p)^{\frac{1}{p}}\right)^p \geq (a_1 + b_1)^p - \Sigma_{i=2}^n (a_i + b_i)^p,$$

*with equality if and only if $a = vb$, where $v$ is a constant.*

**Lemma 2** ([28], p.26). *If $x_i > 0, y_i > 0, i = 1, 2, \cdots, n$, then*

$$\left(\prod_{i=1}^{n}(x_i + y_i)\right)^{\frac{1}{n}} \geq \left(\prod_{i=1}^{n} x_i\right)^{\frac{1}{n}} + \left(\prod_{i=1}^{n} y_i\right)^{\frac{1}{n}}, \tag{2g}$$

*with equality if and only if $\frac{x_1}{y_1} = \frac{x_2}{y_2} = \cdots = \frac{x_n}{y_n}$.*

**Lemma 3** ([5]). *Suppose that $f_i, g_i$ ($i = 1, 2$) are non-negative continuous functions on $\mathcal{S}^{n-1}$ such that*

$$\int_{\mathcal{S}^{n-1}} f_1^s(u)du \geq \int_{\mathcal{S}^{n-1}} f_2^s(u)du,$$

$$\int_{\mathcal{S}^{n-1}} g_1^t(u)du \geq \int_{\mathcal{S}^{n-1}} g_2^t(u)du,$$

*for $s > 1, \frac{1}{s} + \frac{1}{t} = 1$, and*

$$f_1^s(u) = \lambda g_1^t(u), \forall u \in \mathcal{S}^{n-1},$$

*where $\lambda$ is a constant. Then*

$$\left(\int_{\mathcal{S}^{n-1}} (f_1^s - f_2^s)du\right)^{\frac{1}{s}} \left(\int_{\mathcal{S}^{n-1}} (g_1^t - g_2^t)du\right)^{\frac{1}{t}} \leq \int_{\mathcal{S}^{n-1}} (f_1 g_1 - f_2 g_2)du, \tag{2h}$$

*with equality if and only if $f_2^s(u) = \lambda g_2^t(u)$ for any $u \in \mathcal{S}^{n-1}$.*

## 4. Proofs of Main Results

In this section, we prove Theorems 5–7.

**Proof of Theorem 5.** We only prove Equation (1f). The proof of Equation (1g) is similar to Equation (1f). Let $i \leq n - p$. Since $K$ and $L$ have similar general $L_p$-chord, by Equation (1b),

$$C_{p,i}^{(\tau)}(K \bar{+}_p L)^{\frac{p}{n-i}} = C_{p,i}^{(\tau)}(K)^{\frac{p}{n-i}} + C_{p,i}^{(\tau)}(L)^{\frac{p}{n-i}}, \tag{3a}$$

for $M$ and $M'$,

$$C_{p,i}^{(\tau)}(M \bar{+}_p M')^{\frac{p}{n-i}} \leq C_{p,i}^{(\tau)}(M)^{\frac{p}{n-i}} + C_{p,i}^{(\tau)}(M')^{\frac{p}{n-i}}. \tag{3b}$$

Let $a_1 = C_{p,i}^{(\tau)}(K)^{\frac{p}{n-i}}, a_2 = C_{p,i}^{(\tau)}(M)^{\frac{p}{n-i}}$ and $b_1 = C_{p,i}^{(\tau)}(L)^{\frac{p}{n-i}}, b_2 = C_{p,i}^{(\tau)}(M')^{\frac{p}{n-i}}$, then from Equations (3a) and (3b) and Lemma 1, we have

$$\left(C_{p,i}^{(\tau)}(K \bar{+}_p L) - C_{p,i}^{(\tau)}(M \bar{+}_p M')\right)^{\frac{p}{n-i}}$$

$$\geq \left(\left(C_{p,i}^{(\tau)}(K)^{\frac{p}{n-i}} + C_{p,i}^{(\tau)}(L)^{\frac{p}{n-i}}\right)^{\frac{n-i}{p}} + \left(C_{p,i}^{(\tau)}(M)^{\frac{p}{n-i}} + C_{p,i}^{(\tau)}(M')^{\frac{p}{n-i}}\right)^{\frac{n-i}{p}}\right)^{\frac{p}{n-i}}$$

$$\geq \left(C_{p,i}^{(\tau)}(K) - C_{p,i}^{(\tau)}(M)\right)^{\frac{p}{n-i}} + \left(C_{p,i}^{(\tau)}(L) - C_{p,i}^{(\tau)}(M')\right)^{\frac{p}{n-i}}$$

This gives the desired inequality of Equation (1f) and according to the equality condition of Lemma 1, we obtain that equality holds if and only if $M$ and $M'$ have a similar general $L_p$-chord.　□

Notice that from the notion of $L_p$-radial Minkowski homomorphism and Equation (2e), we have the following direct Corollary 1.

**Corollary 1.** *Let $K, L, M, M' \in S_o^n$ and $\tau \in (-1, 1)$, $p > 0$. $\Psi_p$ is a radial Blaschke–Minkowski homomorphism. $K$ and $L$ have a similar general $L_p$-chord and $M \subseteq K$, $M' \subseteq L$, then for $i \leq n - p$,*

$$[C_{p,i}^{(\tau)}(\Psi_p(K \check{+}_p L)) - C_{p,i}^{(\tau)}(\Psi_p(M \bar{+}_p M'))]^{\frac{p}{n-i}} \geq [C_{p,i}^{(\tau)}(\Psi_p K) - C_{p,i}^{(\tau)}(\Psi_p M)]^{\frac{p}{n-i}} + [C_{p,i}^{(\tau)}(\Psi_p L) - C_{p,i}^{(\tau)}(\Psi_p M')]^{\frac{p}{n-i}},$$

*and for $n - p < i < n$ or $i > n$,*

$$[C_{p,i}^{(\tau)}(\Psi_p(K\widetilde{+}_pL)) - C_{p,i}^{(\tau)}(\Psi_p(M\widetilde{+}_pM'))]^{\frac{p}{n-i}} \leq [C_{p,i}^{(\tau)}(\Psi_pK) - C_{p,i}^{(\tau)}(\Psi_pM)]^{\frac{p}{n-i}} + [C_{p,i}^{(\tau)}(\Psi_pL) - C_{p,i}^{(\tau)}(\Psi_pM')]^{\frac{p}{n-i}},$$

*with equality in each inequality if and only if $M$ and $M'$ have a similar general $L_p$-chord.*

Further, since the $L_p$ intersection map is a special $L_p$-radial Minkowski homomorphism, we have the following corollary

**Corollary 2.** *Let $K, L, M, M' \in S_o^n$ and $\tau \in (-1, 1)$, $p > 0$. If $K$ and $L$ have a similar general $L_p$-chord and $M \subseteq K$, $M' \subseteq L$, then for $i \leq n - p$,*

$$[C_{p,i}^{(\tau)}(I_p(K\widetilde{+}_pL)) - C_{p,i}^{(\tau)}(I_p(M\widetilde{+}_pM'))]^{\frac{p}{n-i}} \geq [C_{p,i}^{(\tau)}(I_pK) - C_{p,i}^{(\tau)}(I_pM)]^{\frac{p}{n-i}} + \mu[C_{p,i}^{(\tau)}(I_pL) - C_{p,i}^{(\tau)}(I_pM')]^{\frac{p}{n-i}},$$

*and for $n - p < i < n$ or $i > n$,*

$$[C_{p,i}^{(\tau)}(I_p(K\widetilde{+}_pL)) - C_{p,i}^{(\tau)}(I_p(M\widetilde{+}_pM'))]^{\frac{p}{n-i}} \leq [C_{p,i}^{(\tau)}(I_pK) - C_{p,i}^{(\tau)}(I_pM)]^{\frac{p}{n-i}} + [C_{p,i}^{(\tau)}(I_pL) - C_{p,i}^{(\tau)}(I_pM')]^{\frac{p}{n-i}},$$

*with equality in each inequality if and only if $M$ and $M'$ have a similar general $L_p$-chord.*

**Proof of Theorem 6.** Since $K_1, \cdots, K_n$ have a similar general $L_p$-chord, from (1d) we have for $1 < m \leq n$,

$$C_p^{(\tau)}(K_1, \cdots, K_n)^m = \prod_{i=1}^m C_p^{(\tau)}(K_1, \cdots, K_{n-m}, K_{n-i+1}, K_{n-i+1}, \cdots, K_{n-i+1}). \tag{3c}$$

For $M_1, \cdots, M_n$,

$$C_p^{(\tau)}(M_1, \cdots, M_n)^m \leq \prod_{i=1}^m C_p^{(\tau)}(M_1, \cdots, M_{n-m}, M_{n-i+1}, M_{n-i+1}, \cdots, M_{n-i+1}). \tag{3d}$$

The condition $M_i \subseteq K_i, i = 1, 2, \cdots, n$ means that $C_p^{(\tau)}(K_1, \cdots, K_n)^m \geq C_p^{(\tau)}(M_1, \cdots, M_n)^m$. From Equations (3c) and (3d) and Lemma 2, we obtain

$$C_p^{(\tau)}(K_1, \cdots, K_n) - C_p^{(\tau)}(M_1, \cdots, M_n)$$
$$\geq \left( \prod_{i=1}^m C_p^{(\tau)}(K_1, \cdots, K_{n-m}, K_{n-i+1}, K_{n-i+1}, \cdots, K_{n-i+1}) \right)^{\frac{1}{m}}$$
$$- \left( \prod_{i=1}^m C_p^{(\tau)}(M_1, \cdots, M_{n-m}, M_{n-i+1}, M_{n-i+1}, \cdots, M_{n-i+1}) \right)^{\frac{1}{m}}.$$

Let $x_i + y_i = C_p^{(\tau)}(K_1, \cdots, K_{n-m}, K_{n-i+1}, K_{n-i+1}, \cdots, K_{n-i+1})$ and $y_i = C_p^{(\tau)}(M_1, \cdots, M_{n-m}, M_{n-i+1}, M_{n-i+1}, \cdots, M_{n-i+1})$ in Lemma 2. Then by Equation (2g)

$$C_p^{(\tau)}(K_1, \cdots, K_n) - C_p^{(\tau)}(M_1, \cdots, M_n)$$
$$\geq \left( \prod_{i=1}^m [C_p^{(\tau)}(K_1, \cdots, K_{n-m}, K_{n-i+1}, \cdots, K_{n-i+1}) - C_p^{(\tau)}(M_1, \cdots, M_{n-m}, M_{n-i+1}, \cdots, M_{n-i+1})] \right)^{\frac{1}{m}},$$

which implies that Equation (1h) is proved. According to the equality condition of Lemma 2, we know that equality holds in Equation (1h) if and only if $M_1, \cdots M_n$ all have a similar general $L_p$-chord. $\square$

**Proof of Theorem 7.** For $i < j < k$, let $s = \frac{k-i}{k-j}, t = \frac{k-i}{j-i}$. Then, $s > 1$ and $\frac{1}{s} + \frac{1}{t} = 1$. Let

$$f_1^s = c_p^{(\tau)}(K, u)^{n-i} c_p^{(\tau)}(L, u)^i, \quad f_2^s = c_p^{(\tau)}(M, u)^{n-i} c_p^{(\tau)}(M', u)^i$$

and

$$g_1^t = c_p^{(\tau)}(K,u)^{n-k}c_p^{(\tau)}(L,u)^k, \;\; g_2^t = c_p^{(\tau)}(M,u)^{n-k}c_p^{(\tau)}(M',u)^k.$$

After a simple calculation, we obtain

$$\int_{\mathcal{S}^{n-1}} (f_1g_1 - f_2g_2)du = \int_{\mathcal{S}^{n-1}} (c_p^{(\tau)}(K,u)^{n-j}c_p^{(\tau)}(L,u)^j - c_p^{(\tau)}(M,u)^{n-j}c_p^{(\tau)}(M',u)^j)du$$
$$= C_{p,j}^{(\tau)}(K,L) - C_{p,j}^{(\tau)}(M,M').$$

The left-hand side of Equation (2h) leads to $[C_{p,i}^{(\tau)}(K,L) - C_{p,i}^{(\tau)}(M,M')]^{\frac{1}{s}}[C_{p,k}^{(\tau)}(K,L) - C_{p,k}^{(\tau)}(M,M')]^{\frac{1}{t}}$.

By Lemma 3, Equation (1i) immediately holds.

The equality condition of Equation (2h) means that $\frac{f_1^s}{g_1^t} = \left(\frac{c_p^{(\tau)}(K,u)}{c_p^{(\tau)}(L,u)}\right)^{k-i}$ is a constant, that is, $K$ and $L$ have a similar general $L_p$-chord. This completes the proof. $\quad\square$

## 5. Conclusions

The asymmetric operators belong to a new and rapidly evolving asymmetric $L_p$-Brunn–Minkowski theory that has its origins in the work of Ludwig, Haberl and Schuster (see [9,11,12,16,18–20]). The general $L_p$-mixed chord integral difference of star bodies was motivated by the notion of mixed width-integrals of convex bodies. We hope that besides the inequalities mentioned in this article, we can deduce some other inequalities in the future.

**Author Contributions:** All authors contributed equally and significantly in writing this article. All authors have read and agreed to the published version of the manuscript.

**Funding:** Supported by the Open Research Fund of Computational physics Key Laboratory of Sichuan province, Yibin University: ybxyjswl-zd-2020-004.

**Institutional Review Board Statement:** Not applicable.

**Informed Consent Statement:** Not applicable.

**Data Availability Statement:** Not applicable.

**Conflicts of Interest:** The authors declare no conflict of interest.

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
