# Peer review of "Inequalities on General Lp-Mixed Chord Integral Difference"

_axioms, doi:10.3390/axioms10030220_

Round 1
Reviewer 1 Report
The authors introduce the concept of general $L_p-$mixed chord integral difference of star bodies. Additionally, they prove three results, which involve the Brunn-Minkowski type, Aleksandrov-Fenchel type and cyclic inequalities for $L_p-$mixed chord integral difference.
The three theorems presented in 1.1 to 1.3 and the two corollaries are results that could be interesting from the theoretical point of view, their proof is clear and I found no errors in them.
I have no suggestions.
Author Response
Thank you for your attention on our paper.
Reviewer 2 Report
I have attached the suggestions.

Author Response
Dear Editor and Reviewer,
We would like to thank you for giving us a chance to revise our manuscript, and also thank the reviewers for giving us useful comments and suggestions on our manuscript.
The comments of reviewer 1 mainly focus on spelling errors and nonstandard expressions in English. So accordingly to these comments, we have modified our manuscript one-by-one carefully.
If you have any question about our manuscript, please do not hesitate to let us know.
Thank you again for your attention on our paper.
Yours sincerely,
Zhaofeng Li
Reviewer 3 Report
please see the attachment.

Round 2
Reviewer 3 Report
please see the report attached.

Round 3
Reviewer 3 Report
I am happy to recommend the paper for publication now.